# To Work at Clinics or at Hospitals? Analysis of Family Physician Recruitment Advertising in Taiwan

**DOI:** 10.3390/ijerph17165636

**Published:** 2020-08-05

**Authors:** Yi-Shin Cheng, Ann Charis Tan, Ya-Chuan Hsu, Tzeng-Ji Chen

**Affiliations:** 1Department of Family Medicine, Taipei Veterans General Hospital, No. 201, Sec. 2, Shi-Pai Road, Taipei 112, Taiwan; ffaiai107@gmail.com (Y.-S.C.); ych97160@gmail.com (Y.-C.H.); 2Department of Medicine, Taipei Veterans General Hospital, No. 201, Sec. 2, Shi-Pai Road, Taipei 112, Taiwan; actan5@gmail.com; 3School of Medicine, National Yang-Ming University, No. 155, Sec. 2, Linong Street, Taipei 112, Taiwan; 4Big Data Center, Department of Medical Research, Taipei Veterans General Hospital, No. 201, Sec. 2, Shi-Pai Road, Taipei 112, Taiwan

**Keywords:** advertising, family practice, health workforce, periodicals as topic, personnel selection, supply and distribution, Taiwan

## Abstract

Family physicians act as gatekeepers of the healthcare system and have an indispensable role in providing holistic care in the primary care system. While previous studies had focused on the geographic maldistribution of family physicians, the current study investigated the distribution of job opportunities for family physicians by analyzing recruitment advertisements posted in medical association journals, as an indirect way to observe the marketplace demand for physicians. We collected all the recruitment advertisements for family physicians in the twelve issues of the Taiwan Medical Journal, the official organ of the Taiwan Medical Association, in 2018. In contrast to 124 new trainees annually, 739 advertisements for family physicians were posted within the entire year. After eliminating repeated advertisements, there were 302 distinct advertisements, of which hospitals accounted for 18.9% (*n* = 57). The job opportunities at hospitals were offered mainly by regional hospitals (*n* = 26) and community hospitals (*n* = 29), but only two by medical centers. Family physicians in Taiwan were in great demand not only by primary care clinics but also by hospitals. The role of family physicians in hospitals is worth further study.

## 1. Introduction

Family medicine is a specialty with the most versatility among all the medical specialties, providing comprehensive medical care to patients of all ages and genders. Family physicians are just as diverse as their patients and are known to have much flexibility in the choice of practicing. Compared with other specialties that often need advanced equipment which can only be offered in hospitals, family physicians are more likely to practice in diverse locations, including the rural or countryside areas. However, previous studies have discussed the phenomenon of geographic maldistribution of primary care physicians in the United States and in Japan, showing the dearth of family physicians in rural areas [1,2,3]. A 2018 statistics report in Taiwan showed that about 70% of family physicians work in metropolises, which account for 30% of Taiwan’s land area in 2018 [4]. 

Aside from collecting data by questionnaires to evaluate the subjective factors for career choice, some studies also analyze the need for family physicians by observing recruitment advertisements posted in medical journals. The latter method is more objective in describing the social recognition of family physicians [5]. The changing number of recruitment advertisements provides an idea of the change of marketplace demand for physicians where the ongoing analysis of advertisements provides timely information about the demand for physicians in a rapidly changing health care system. Previous studies had discussed about the recruitment of different specialties including radiation oncology [6], radiology [7], dermatology [8], dentistry [9], and general medicine [10]. In the study of the radiation oncology job market, job availability differed among regions, and more open positions were in rural locations [6]. A help-wanted index was used in the detecting of an interventional radiologist job market, revealing a dramatic job shift [7]. With regards to the field of dermatology, a rising shortage of dermatologists and an increase in demand for dermatologic services were noted [8], so as the finding in the dental faculty [9]. A similar predicament was revealed in general physicians [1]. Due to the general physician shortage, most family physician-related studies focused on strategies to recruit and retain primary care doctors [10,11,12].

Within the healthcare system and National Health Insurance program in Taiwan, patients can visit specialists of any type at any level of facilities directly without referrals. Inevitably, the role of family physicians as gatekeepers is weakened and hospitals take responsibility for primary health care. Therefore, many family physicians choose to remain to work in hospitals (e.g., medical center, regional hospital, and community hospital) rather than in rural clinics [12]. The Taiwan government had been promoting a Family Physician Integrated Care Project since 2003 to highlight community medical group-based practice [13].

The current study aimed to analyze the nationwide recruitment of family physicians in Taiwan in 2018, stratified by area and type of medical facilities. Furthermore, the focus would be on situations in hospitals, especially the workforce distribution and need for family physicians at different levels of hospitals under the current health system and policy.

## 2. Materials and Methods 

### 2.1. Background

Taiwan is a country with 23 million people, located at the eastern part of Asia. The Taiwan Medical Association was established in 1930, and there were over 49 thousand members in 2018, practicing in 22 medical specialties [14]. Family medicine, which topped the list of the specialties announced by the Ministry of Health and Welfare, was introduced to provide holistic care and to act as a gatekeeper of the health care system [15]. The Association of Family Medicine was formally established in 1986 in Taiwan to promote family medicine research and development, to develop a family medicine specialist system in Taiwan, to strengthen contact and exchange with international family medicine organizations, and to raise primary medical care and family medicine standards [16]. In 2018, there were 3655 family medicine practitioners working in different levels of medical facilities, including residents who were training under the family medicine specialty [17].

After six years of studying in medical schools and five years working as residents, there are approximately 120 newly trained family medicine practitioners each year in Taiwan, and most of them work as general practitioners after obtaining a specialist license, either in hospitals or in local clinics [18]. 

### 2.2. Data Source and Study Design

We collected all the recruitment advertisements in the twelve issues of the Taiwan Medical Journal in 2018, which were published by the Taiwan Medical Association every month since 1957, targeted for all practitioners in Taiwan, with the content of updating medical information of various specialties. We recorded the advertisements on family physician recruitment and marked the advertisements that were repeatedly posted. We also recorded the advertisements of each medical care facility with their location and accreditation level. The numbers of medical care facilities and practitioners presented were from the statistics of the Taiwan Medical Association. The data of land area and population were from the statistics of Taiwan’s Ministry of the Interior.

We collected the information of all the advertisements, including the name of the facility, location, specialty, number of physicians required, and special requirements. There were some advertisements that did not show the exact number of physicians needed, and we counted them as one recruitment each. Some facilities have different branches in different cities, and we counted them as independent facilities separately. There were repeatedly posted advertisements. We marked them and compared the number between newly posted and repeatedly posted advertisements in each month. 

### 2.3. Implementation

The data collected in the twelve issues of the Taiwan Medical Journal were first grouped into two groups—hospital and clinic—and the data in the hospital group were then further divided into three groups—medical center, regional hospital, and community hospital. To observe the difference between different locations, the data were divided into 7 groups according to their location: northern Taiwan consists of Taipei City, New Taipei City, and Keelung City; northwest Taiwan consists of Taoyuan City, Hsinchu County, Hsinchu City, and Miaoli County; central Taiwan consists of Taichung City, Changhua County, and Nantou County; southwest Taiwan consists of Yunlin County, Chiayi County, Chiayi City, and Tainan City; southern Taiwan consists of Kaohsiung City and Pingtung City; eastern Taiwan consists of Yilan County, Hualien County, and Taitung County; and offshore islands consist of Lienchiang County, Kinmen County, and Penghu County. We also surveyed the population density, dependency ratio, number of family physicians, number of family medicine residents, and number of family physician training facilities of each location. We considered that population and age structure might contribute to the demand for family physicians under the presumption that each family physician can serve the same number of patients.

We surveyed the number of practitioners, training residents, and recruitments. The number of practitioners was calculated as the number of family medicine members minus the number of residents, with data collected from the Taiwan Medical Association and the Taiwan Association of Family Medicine, respectively.

### 2.4. Statistical Analysis

The data collection and analysis were performed with Microsoft Excel 2016 (Redmond, Washington, DC, USA) and presented in descriptive statistics. The categorical variables were presented in numbers and percentages.

## 3. Results

### 3.1. General Findings

There was a total of 12 Taiwan Medical Journal issues published in 2018, one for each month, and an overall number of 739 recruitment advertisements for family physicians were posted in these issues. There were 312 advertisements posted by hospitals and 427 advertisements by clinics, and the number varied in each month (Figure 1). The advertisements were further grouped into newly posted and repeatedly posted ones. For the entire year of 2018, hospitals posted 57 new and 255 repeat advertisements, while clinics posted 245 new and 182 repeat advertisements.

### 3.2. Diversity and Trend of Recruiting Advertisements for Family Physicians in Each Month of 2018

There were more advertisements posted by clinics from September to December 2018 wherein the combined number of newly posted and repeatedly posted advertisements reached more than 40 per month. In contrast, the advertisements posted by hospitals remained mostly the same, with the combined number of advertisements around 20 to 30 per month. It was notable that for every month in 2018, clinics consistently posted more newly posted recruitment advertisements than hospitals (Figure 1).

### 3.3. Distribution of Family Physicians and Recruitments by Area in Taiwan

General information about each area is shown in Table 1 including population density, dependency ratio, number of family physicians, number of family medicine residents, number of family physician training facilities, and number of recruitment advertisements. With regards to the geographic distribution of family physicians and recruitments in Taiwan, central Taiwan had the largest number of recruitment advertisements (276 out of 739), as well as distinct recruitments (94 out of 302) and ratio of recruitment over physician (42.9%). Northwest Taiwan (141, 32.1%) and southwest Taiwan (108, 21.2%) were the second and third most in number and ratio of recruitment over physician. The smallest numbers of recruitments were from eastern Taiwan (3) and offshore islands (0) (Table 1).

### 3.4. Distribution of Family Physicians and Recruitments by Accreditation Level of Medical Care Facility

With regard to the distribution of family physicians and recruitments by accreditation level of the medical care facility, there were 2143 (67.9%) physicians practicing in clinics and 1011 (32.1%) physicians practicing in hospitals, with 461 (14.6%) in regional hospitals, 333 (10.6%) in community hospitals, and 217 (6.9%) in medical centers. There were 245 (81.1%) distinct recruitments posted by clinics and 57 (18.9%) recruitments posted by hospitals, with 29 (9.6%) by community hospitals, 26 (8.6%) by regional hospitals, and 2 (0.7%) by medical centers. The ratio of recruitment over physician was 11.4% in clinics, 8.7% in community hospitals, 5.6% in regional hospitals, and 0.9% in medical centers. Most residents were trained in medical centers (296, 59.1%), followed by regional hospitals (186, 37.1%) and community hospitals (19, 3.8%) (Table 2).

## 4. Discussion

Our study investigated the factors that influence family physician recruitment conditions in Taiwan, specifically through recruitment advertisements. We found that the number of advertisements posted by clinics was more than those posted by hospitals, and the numbers differed by month, location, and accreditation level of the medical facilities. 

The number of advertisements posted by clinics surpassed those by hospitals, especially in September to December 2018. This finding may be correlated with the schedule of the family medicine specialist examination where the exam is held in November and December every year, starting with a written exam and followed by an oral exam two weeks later [25]. During this period, most residents have finished the required training course in hospitals and would search for opportunities in various medical facilities. This would influence the recruitment condition and turnover of the job market. However, this phenomenon was not reflected in the recruitment advertisements posted by hospitals. Previous studies showed that family physicians working in hospitals were significantly more likely than their non-hospitalist peers to work longer hours, have better pay, and be more satisfied with their work [26,27]. These attributes may lead to a low turnover rate in hospitals.

Northern Taiwan is an area in Taiwan with the largest and densest population in the nation, and central Taiwan has about two-thirds of the population of northern Taiwan. A larger population would presumably present a larger demand for physicians. However, findings revealed that the number of recruitment advertisements in central Taiwan was twice that in northern Taiwan. This indicated that the turnover rate in central Taiwan was higher regardless of its small population. One reason may be that the number of medical centers differed in these two areas. There are 12 medical centers in northern Taiwan, but only 6 in central Taiwan, since most hospitals in that area were either regional or community hospitals. The high turnover rate in central Taiwan may indicate that physicians tend to search for new jobs after residency training instead of staying in the hospitals where they were initially trained in [28,29].

In Taiwan, only 67.9% of family physicians work in local clinics. However, most recruitment advertisements (81%) were posted by local clinics. This showed that the supply was less than the demand. Another reason that may be attributed to this finding is that because all residents were trained in hospitals, there would be a certain number of physicians who chose to stay in the same hospitals they were trained in without the help of recruitment advertisements [30]. In this study, we observed that the majority of residents were trained in medical centers (59.1%) as opposed to regional or community hospitals. This may be the reason why the number of advertisements posted by medical centers (2, 0.7%) was less than that posted by regional hospitals (26, 8.6%) and community hospitals (29, 9.6%). The supply was more than the demand in medical centers. Therefore, advertisements were not the main source of recruitment. On the other hand, there were about 124 new trainees annually, and the demand shown by recruitment advertisements by hospitals accounted for nearly half of the numbers despite there being other sources of recruitment. The roles of family physicians in hospitals are worth further research [31].

Overall, this study presented the annual family physician demand for different months, different localities, and different accreditation levels of the medical care facility. This corresponds to the fact that family physicians in Taiwan not only provide frontline medical service in primary care facilities but also play an indispensable role in hospitals. Most residents were trained in hospitals, and they tend to stay in the same hospital after finishing training. Nevertheless, hospitals still need to post recruitment advertisements because the lack of family physicians in hospitals still exists. This might be related to the additional workload of family physicians aside from outpatient service, where they would need to promote medical screening for at-risk groups and vaccination programs, among others. The work and role of family physicians in hospitals that is related to public health policy development calls for a further study.

### Limitations

The source of recruitment advertisements for our study was limited to the Taiwan Medical Journal. There were other recruitment platforms such as the internet, newspapers, and internal recruitment that were not included. Even with the only journal, our data might be underestimated because some advertisements without specifying the number of physicians needed were operationally deemed as one physician needed. In this study, we counted only the amount of recruitments and did not analyze the working conditions, salaries, and benefits, which were usually not provided in the advertisements. The advertisements may not be limited only to residents, but targeted to family physicians in general. The number of advertisements in an area may also correlate with a high turnover rate where, for example, an area with a higher number of advertisements may not be in a greater shortage of family physicians than other areas. Furthermore, our study described only job offers, but the actual applications and the results remained unknown, which warrants further research for clarification.

## 5. Conclusions

According to our analysis of recruitment advertisements in the Taiwan Medical Journal, family physicians in Taiwan were in great demand not only by primary care clinics, but also by hospitals, especially by regional and community hospitals. Beyond the fact that family physicians provide frontline medical services in the primary care system, there is a need to conduct further studies discussing the roles of family physicians in hospitals and the association between the supply and demand of the family physician workforce.

## Figures and Tables

**Figure 1 ijerph-17-05636-f001:**
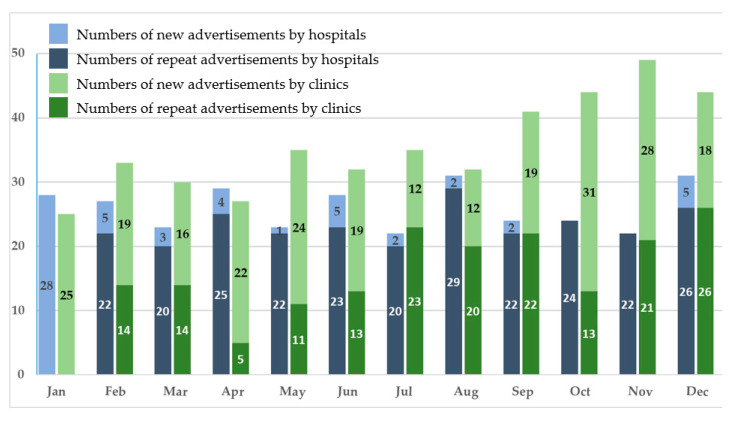
Family physician recruitment advertisements in the 2018 Taiwan Medical Journal.

**Table 1 ijerph-17-05636-t001:** Distribution of family physicians and recruitments by area in Taiwan.

Location	Population Density, km^2^ [18]	Dependency Ratio, % * [18]	Family Physicians, *n* (%) [12]	Family Medicine Residents, *n* (%) [19]	Family Physician Training Facilities, *n* (%) [19]	Total Recruitments, *n* (%)	Distinct Recruitments, *n* (%)	Ratio of Recruitment over Physician, %
North	2863	38.3	821 (26.0)	183 (36.5)	20 (29.9)	129 (17.5)	64 (21.2)	15.7
Northwest	825	37.8	439 (13.9)	32 (6.4)	7 (10.4)	141 (19.1)	51 (16.9)	32.1
Central	619	37.5	643 (20.4)	123 (24.6)	16 (23.9)	276 (37.3)	94 (31.1)	42.9
Southwest	571	38.8	509 (16.1)	78 (15.6)	12 (17.9)	108 (14.6)	55 (18.2)	21.2
South	628	37.0	493 (15.6)	66 (13.2)	7 (10.4)	82 (11.1)	35 (11.6)	16.6
East	97	38.7	208 (6.6)	19 (3.8)	5 (7.5)	3 (0.4)	3 (1.0)	1.4
Offshore Islands	836	31.6	41 (1.3)	0 (0)	0 (0)	0 (0)	0 (0)	0
Total	645	37.9	3154 (100)	501 (100)	67 (100)	739 (100)	302 (100)	23.7

* Dependency ratio: a measure of the number of dependents aged 0 to 14 and over the age of 65, compared with the total population aged 15 to 64, calculated as: (numbers of aged 0 to 14 and over the age of 65)/(numbers of aged 15 to 64).

**Table 2 ijerph-17-05636-t002:** Distribution of family physicians and recruitments by medical care facility accreditation level in Taiwan.

	Medical Center	Regional Hospital	Community Hospital	Clinic	Total
Numbers of medical facilities [20]	25	77	346	11547	11995
Family physicians, *n* (%) [19,21,22,23,24]	217 (6.9)	461 (14.6)	333 (10.6)	2143 (67.9)	3154 (100)
Family medicine residents, *n* (%) [19]	296 (59.1)	186 (37.1)	19 (3.8)	0 (0)	501 (100)
Distinct recruitments, *n* (%)	2 (0.7)	26 (8.6)	29 (9.6)	245 (81.1)	302 (100)
Ratio of recruitment over physician, %	0.9	5.6	8.7	11.4	9.6

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
