# Peer review of "To Work at Clinics or at Hospitals? Analysis of Family Physician Recruitment Advertising in Taiwan"

_ijerph, 2020, doi:10.3390/ijerph17165636_

Round 1

Reviewer 1 Report

This paper provides interesting insight into demands of family physicians in Taiwan, but there are several points which needs reconsideration. Please refer to the text attached.

Author Response

Please see the attachment for the response to the reviewer’s comments. Thank you!

Reviewer 2 Report

1) Although the journal did not strongly recommend a literature review section, a solid literature is highly needed. The reader cannot follow the study as there are no background of the study. 

2) The introduction is too short. The authors should indicate the purpose, significant, and the theory of this study. 

3) How can this research contribute to the current field? 

4) An implication section is strongly needed. 

5) Again, after the reader read the paper multiple times, the reader still cannot indicate any new finding of this study. It is good to review some medical journal related issues. However, the finding did not have any additional inputs for the current society. 

Additional experiments and research activities are needed before it can be published at the SSCI level. 

Author Response

(The authors gave the same response as above.)

Round 2

Reviewer 1 Report

Thank you for your revised version of the manuscript. Although the manuscript has significantly improved, I recommended some corrections.

Author Response

Please see the attachment for the response to reviewer's comment.

Reviewer 2 Report

Thank you for the upgrading. 

I totally agreed that the authors would like to upgrade the papers. However, there are still a large room for improvement. 

1) The reviewer always expected a detailed literature review. The current version reviewed only a few studies. But the number is not enough for a holistic picture of the social issue and problem. It can be good for some readers. But for scientists in other fields, the background is totally not enough. For example, Line 51, the authors grouped [8-11] in the same sentence. The reviewer always expected the expanded version of these literature. For example, how ocnologist [8] said and what were the results of this paper? The author is looking for these results. 

For the SSCI-level paper, this is a need to expand the knowledge to readers without solid background in the field. 

2) The authors need to expand a new section 1.1 Purpose of the study in order to outline and categorise the chapter. 

3) The authors need to write down the research questions and/or hypothesis for the study. Currently, on Chapter 1, it is hard to see what is/are your research questions. I can see your aims, but what are the research questions. 

4) Besides, the authors need to expand the literature review about the current issues in Taiwan. Taiwan has a lot of problems in this field. There is a need to expand the background and literature review coverage in order to make the research sounds. 

5) Line 153, always not to say this is the first study. In the future, if someone discovered that this is not the first study, this study will be withdrawn. 

6) Please add an implementation Section or Chapter. 

Author Response

Please see the attachment for the response to the reviewer's comment.

Round 3

Reviewer 1 Report

Thank you for your revised manuscript. The manuscript has much improved.

Author Response

Thank you very much for all the suggestions to make this paper complete. 

Sincerely yours,

Tzeng-Ji Chen, Dr. med.

Section Editor for 'Health Care Sciences & Services'

Professor

Institute of Hospital and Health Care Administration, School of Medicine, National Yang-Ming University, Taipei, Taiwan

Director

Department of Family Medicine, Taipei Veterans General Hospital, Taipei, Taiwan

(Corresponding Author)

Yi-Shin Cheng, MD

Resident

Department of Family Medicine, Taipei Veterans General Hospital, Taipei, Taiwan

Ann Charis Tan, M.S.

Research Assistant

Department of Medicine, Taipei Veterans General Hospital, Taipei, Taiwan

Ya-Chuan Hsu, MD

Research Fellow

Department of Family Medicine, Taipei Veterans General Hospital, Taipei, Taiwan

Reviewer 2 Report

The authors improved the papers based on the recommendations. 

Several recommendations should be followed:

1) The reviewer is still looking for a longer literature review background as the reviewer and potential readers would not understand without a longer background of the study. The study is meaningful in fact. But the reviewer wants to understand why this project is so significant? 

2) The reviewer has recommended an implementation section. Please add it for the second version. 
